# CAN 3D VISION-LANGUAGE MODELS TRULY UNDER-STAND NATURAL LANGUAGE?

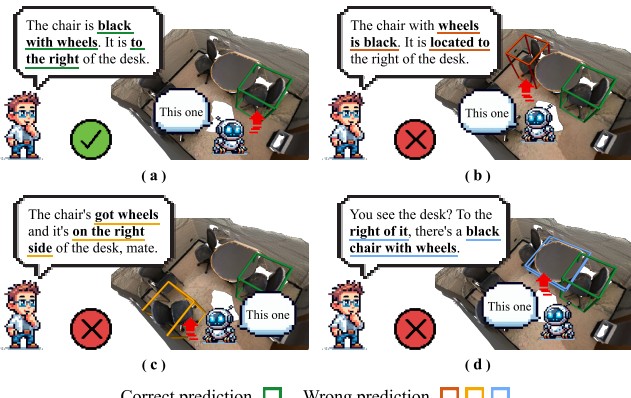

Figure 1: Fragility of 3D-VL models in natural language understanding. This figure shows the failure of 3D-VL models when faced with natural language variations common in human communication. The variations include: *(a)* The original sentence in the training set. *(b)* Shifting the voice from active voice to passive voice. *(c)* Saying the same thing in a different accent. *(d)* Saying in a new conversation tone. Such variations are common in human language, but the model fails at them.

## ABSTRACT

Rapid advancements in 3D vision-language (3D-VL) tasks have opened new avenues for human interaction with embodied agents or robots using natural language. Despite this progress, we find a notable limitation: existing 3D-VL models exhibit sensitivity to the styles of language input, struggling to understand sentences with the same semantic meaning but written in different variants. This observation raises a critical question: **Can 3D vision-language models truly understand natural language?** To test the language understandability of 3D-VL models, we first propose a language robustness task for systematically assessing 3D-VL models across various tasks, benchmarking their performance when presented with different language style variants. Importantly, these variants are commonly encountered in applications that require direct interaction with humans, such as embodied robotics, given the diversity and unpredictability of human language. We propose a 3D Language Robustness Dataset, designed based on the characteristics of human language, to facilitate the systematic study of robustness. Our comprehensive evaluation uncovers a significant drop in the performance of all existing models across various 3D-VL tasks. Even the state-of-the-art 3D-LLM fails to understand some variants of the same sentences. Further in-depth analysis suggests that existing models have a fragile and biased fusion module, which stems from the low diversity of the existing dataset. Finally, we propose a training-free module driven by LLM, which improves language robustness.

## 1 INTRODUCTION

In recent years, the connection of vision and language has attracted considerable interest (Liu et al., 2023; Li et al., 2023). Significant progress has been made on various tasks, such as Visual Grounding and Visual Question Answering (VQA), in the context of understanding both 2D Vision Language (2D-VL) (Tiong et al., 2022; Fukui et al., 2016; Antol et al., 2015; Anderson et al., 2018; Xie et al., 2023; Zhong et al., 2022; Gao et al., 2017) and 3D Vision Language (3D-VL) (Huang et al., 2022; Yang et al., 2021b;a; Ding et al., 2023b; Yang et al., 2022; Ding et al., 2022; 2023a; Yang et al.,

2024b). These tasks represent the foundation skills in real-world applications such as the description of images (Vinyals et al., 2015), embodied robotics (Gupta et al., 2021), AR/VR, and autonomous agents (Xi et al., 2023; Yang et al., 2024a) that require human-machine interaction. They necessitate the model's ability to comprehend free-form natural language instructions for generating predictions.

2D-VL models can handle various prompts (Lai et al., 2023), benefiting from large-scale and diverse Internet-sourced image-language datasets as shown in Fig. 2(c). These datasets contain a wide range of natural language expressions, which enhances the robustness of 2D-VL towards different language styles. However, we did not observe the same success in the 3D-VL domain. Instead, we observe a notable limitation: Existing 3D-VL models exhibit **bias towards language styles in their training datasets** and **struggle to understand minor variations** in our daily languages. As shown in Fig. 1, even minor variations in expression conveying the same meaning can result in model failure. Understanding different language styles is crucial for real-world applications, such as embodied robotics, where humans tend to use a variety of expressions instead of adhering to a fixed language pattern (Holtzman et al., 2019).

Unfortunately, replicating the success of 2D-VL by obtaining large and diverse 3D-VL datasets is both challenging and resource-intensive, hindering the success of 3D vision-language tasks compared to their 2D counterparts. This disparity in dataset diversity and robustness motivates us to systematically study the language robustness of 3D-VL models and explore methods to improve them without relying on extensive datasets. Currently, it lacks a suitable task or dataset designed to facilitate this line of study. Recent studies have assessed the robustness of 2D-VL models using negative samples such as semantically altered instructions (Yuksekgonul et al., 2022; Hendricks & Nematzadeh, 2021; Wang et al., 2023a; Thrush et al., 2022; Zhao et al., 2022) and evaluated the resilience of LLMs to typo errors (Liang et al., 2022) (semantic preservation), which diverges from our research focus. Semantic alterations compromise the objectives of grounding or QA by distorting meaning, while simple typos fail to capture the systematic diversity of human language. Besides, we focus on studying model robustness toward **natural variations** of sentences without altering their meanings, which is more practical for real-world applications in embodied agents and robotics.

Thus, we introduce the 3D Language Robustness (3D-LR) Benchmark, designed for a comprehensive evaluation of the language robustness in 3D-VL models. Specifically, our benchmark evaluates various 3D-VL models on different tasks (Achlioptas et al., 2020; Yang et al., 2021b; Chen et al., 2020; Huang et al., 2022; Azuma et al., 2022), utilizing a specially curated 3D Language Robustness dataset. This dataset challenges models with a variety of language style variants. To accurately model human natural language, we first identify the five most common key language styles variants of natural language in human communications: syntax, voice, modifier, accent, and tone, drawing from established linguistic theories (Barber et al., 2009; Bhagat & Hovy, 2013). Each variant corresponds to a specific aspect of language commonly used in human communication. For example, syntax involves altering sentence structures, while voice entails transitioning between active and passive forms. (More details will be shown in Sec. 3) Subsequently, we developed a paraphrasing pipeline leveraging a Large Language Model (Raffel et al., 2020; Brown et al., 2020; Wei et al., 2021; Sanh et al., 2021; Ouyang et al., 2022) (LLM) to generate the 3D-LR dataset. This process involves rewriting sentences from existing 3D-VL (Achlioptas et al., 2020; Chen et al., 2020; Azuma et al., 2022) datasets. We prompt the LLM with strict rules and linguistic theory, instructing it to rephrase sentences into designated language styles while preserving their original meaning. Furthermore, we employ both statistical analyses and neural-based semantic quality assessments to verify that the variants preserve the same meaning. The evaluation results reveal that even state-of-the-art methods struggle with minor sentence style changes, experiencing performance decreases of up to 32%. Notably, powerful LLM-based methods, such as 3D-LLM (Hong et al., 2023), whether fine-tuned with task-specific supervision or not, also exhibit performance degradation.

Apart from our 3D-LR benchmark, we further propose a simple LLM-based pre-alignment module that enhances robustness in 3D-VL models without additional training. Our method successfully narrows the performance gap by up to 80%. Remarkably, it performs as well as models augmented with double the training data size (from 40k to 80k). Through our comprehensive analysis, we have identified that the fusion module in existing models acts as a major point of failure, as it is biased toward the training dataset. Our proposed method can effectively address this issue.

In summary, our primary contributions are:

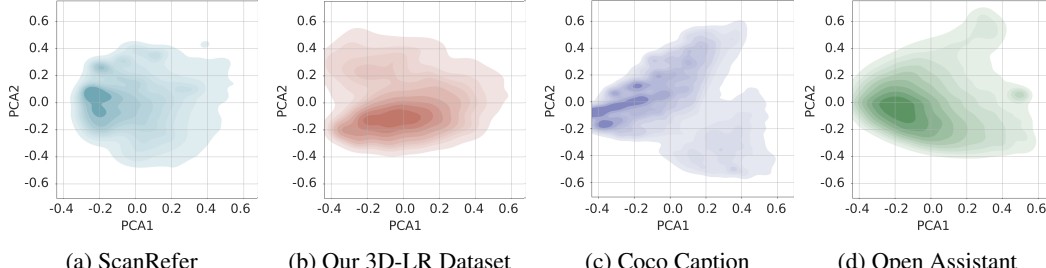

(a) ScanRefer      (b) Our 3D-LR Dataset      (c) Coco Caption      (d) Open Assistant

Figure 2: Density map of four datasets' vectorized syntax structure principal features. Darker areas indicate a higher density of similar sentence patterns, suggesting the dataset contains simple and less diverse structures. (a) ScanRefer (Chen et al., 2020). (b) Our 3D-LR Dataset. (c) Coco Caption. (d) Open Assistant (Köpf et al., 2023). More details in Suppl.

1) We study language robustness toward natural variations of sentences without altering their meanings, which is more practical for real-world applications. We aim to answer an important question: **Can 3D vision-language models truly understand natural language?**

2) We conduct a systematically designed 3D language robustness dataset based on linguistic theories, which properly models real-world natural language to facilitate system benchmarking. Our benchmarks on various 3D-VL models revealed their vulnerability to language patterns. Further in-depth analysis showed that this issue stems from the fusion module, primarily caused by the limited diversity in the training datasets.

3) We propose a simple yet effective training-free LLM-based pre-alignment module that can recover a large proportion of performance without training.

## 2 RELATED WORKS

**3D Vision-Language (3D-VL).** 3D-VL understanding tasks, such as 3D Visual Question Answering (3D-VQA) and 3D Visual Grounding (3D-VG), are pivotal for embodied agents and robotics. These tasks require the simultaneous perception of the 3D world and an understanding of natural language. In 3D-VQA, models select the correct answer from a set of candidates based on the input 3D scene and a natural language question (Azuma et al., 2022) while 3D-VG involves selecting the correct object (Achlioptas et al., 2020; Chen et al., 2020).

The key to both tasks is aligning 3D and text, leading them to have similar model architectures. The most common architectural design is a dual streams model (Azuma et al., 2022; Zhang et al., 2023; Ye et al., 2022; Ma et al., 2022; Achlioptas et al., 2020; Abdelreheem et al., 2022; Chen et al., 2020). This comprises a visual encoder (Qi et al., 2017a), and a text encoder (LSTM (Hochreiter & Schmidhuber, 1997) or BERT (Devlin et al., 2019)) that converts natural language. Then, a fusion module aligns features before passing them to the prediction head. Additionally, leveraging multi-view 2D images has shown promise in enhancing performance (MVT (Huang et al., 2022), SAT (Yang et al., 2021b)). Inspired by the 2D vision large language model pre-training, there are some attempts at bringing LLM into 3D-VL (Hong et al., 2023). This line of work adapts 3D features into language space, enabling the LLM to use visual features as conditions to make predictions.

Existing 3D-VL datasets are either built by programs or human (Achlioptas et al., 2020; Chen et al., 2020). There is debate about whether these datasets accurately model 3D-VL tasks. SQA3D (Ma et al., 2022) highlights an embodied agent's need to know its situation and introduces a dataset with situation descriptions. Multi3Drefer (Zhang et al., 2023) challenges the assumption in existing 3D-VG tasks that only one object is referenced in a sentence. Despite these advancements, a noticeable "domain gap" exists between dataset language and everyday human language. Even human-labeled datasets lack linguistic diversity, exhibiting a fixed pattern (Fig. 2(a)) distinct from the varied expressions of natural communication (Fig. 2(d)). We therefore propose a language robustness task with an evaluation dataset to assess models on varied language styles.

**Model Robustness Studies.** Several studies have explored visual robustness in 2D (Hendrycks & Dietterich, 2019) and 3D (Ren et al., 2022). Language robustness research falls into two categories: semantic alteration and semantic preservation. 1) **Semantic alteration** modifies textual elements (e.g., rearranging phrases, swapping attributes) (Yuksekgonul et al., 2022; Hendricks &

Nematzadeh, 2021; Thrush et al., 2022; Zhao et al., 2022) to test model behavior. However, this approach is unsuitable for 3D-VL tasks like 3D-VG and 3D-VQA, which require high semantic accuracy. Altering meaning contradicts these tasks' goals. We focus on natural human expression diversity rather than creating unnatural text. 2) **Semantic preservation** examines how LLMs handle minor typos (Liang et al., 2022), but this work is confined to unimodal NLP and overlooks complex linguistic variations. Our work belongs to semantic preservation but moves beyond simple typos to investigate complex, systematic, and practical natural language variations.

## 3  3D LANGUAGE ROBUSTNESS (3D-LR) BENCHMARK

To systematically evaluate the fragility of 3D-VL methods to various language styles, we propose the 3D Language Robustness benchmark. The subsequent sections detail our benchmark design, starting with our proposed 3D Language Robustness Task in Sec. 3.1. Following this, we introduce our 3D Language Robustness dataset (3D-LR) in Sec. 3.2, including its design principles, construction pipeline, and key statistics.

### 3.1  3D LANGUAGE ROBUSTNESS TASK

Motivated by the above, we present the 3D Language Robustness task. This task is designed to evaluate the generalization capabilities of a pre-trained 3D-VL model across diverse language variants of a given dataset. Specifically, this evaluates the model's ability to understand and process sentences that have the same meaning as the original but are expressed differently.

Formally, a standard 3D-VL task such as 3D-VQA (Azuma et al., 2022), 3D-VG (Chen et al., 2020) can be viewed as the model takes two modality inputs: a 3D scene represent in a $k$ points point cloud $\mathcal{P} = \{(p_i, f_i); i = 1, 2, \ldots, k\}$, where $p_i \in \mathcal{R}^3$ denotes the coordinates and $f_i$ represents extra features and a natural language sentence $\mathcal{S} = [s_0, s_1, ..., s_n]$ representing a free-from sentence with $n$ words. The model treats these tasks as a classification problem over a predefined set of candidate answers or objects.

To simulate daily natural language variants, we first derived the five most representative language characteristics humans use in communication from linguistic theory (Barber et al., 2009; Bhagat & Hovy, 2013). Based on these characteristics, we design five rephrasing operators $o \in \{N = 5 \text{ styles}\}$ to translate the original 3D-VQA and 3D-VG datasets, $\mathcal{D}$, into various styles while maintaining the meaning. Resulting in different sets of dataset splits, denoted as $\mathcal{D}_o$, containing sentence variants respectively. These different language variants contain the same semantics as the original sentences. They include the very same keywords. In other words, all language variants share the same language clues to complete the task. Given such a set of textual variant splits has the same meaning. A model, denoted as $\mathcal{M}$ trained on the original data $\mathcal{D}$ is evaluated on the newly built data splits $\mathcal{D}_o$ as described above. We have five language variant splits in our setting. Aiming to evaluate the performance degradation comprehensively.

### 3.2  3D LANGUAGE ROBUSTNESS (3D-LR) DATASET

**Background of Language Characteristics.** Human language conveys meaning through its flexible syntax, grammar structures, and features like voice and tones. We use **five** main characteristics to model human language inspired by linguist theories (Barber et al., 2009; Bhagat & Hovy, 2013). Firstly, **Syntax** refers to varying word or phrase orders to create different sentence structures; for example, inverse sentences are commonly used in daily conversation. Secondly, **Voice** involves paraphrasing a sentence from active to passive voice, or vice versa, a fundamental aspect of language. Thirdly, **Modifier**, such as adjectives and adverbs, are varied by humans to enhance the details in a sentence, adding richness and depth to communication. Fourthly, **Accent** reflects the distinct linguistic habits of English speakers from different regions, characterized by unique vocabulary and sentence structures. These regional variations, however, do not change the fundamental meaning of the communication. Finally, **Tone** encompasses the attitudes and emotions conveyed in a sentence, which vary across different contexts. An example is the use of questions in daily conversation, which demonstrates how tone can add layers of meaning beyond the literal interpretation of words. Some examples are shown in Fig. 3. A detailed explanation is shown in the supplementary file.

**LLM Rephrasing for Dataset Construction.**  We aim to create a dataset including the language variants defined earlier to systematically assess the language robustness of existing models across different linguist styles.  Our approach involves rephrasing sentences from existing

Table 1: Dataset quality assessment and basic statistics of our *ScanRefer-R*.

**(a)** Semantic similarity between *ScanRefer-R* and the original dataset.

|  | BERT Sim. (↑) | Glove Sim. (↑) | ED (↓) |
|---|---|---|---|
| Syntax | 0.92 | 0.98 | 0.42 |
| Voice | 0.92 | 0.98 | 0.60 |
| Modifier | 0.91 | 0.96 | 0.51 |
| Accent | 0.83 | 0.89 | 0.63 |
| Tone | 0.86 | 0.93 | 0.51 |

**(b)** Basic statistics of different data splits in *ScanRefer-R*.

|  | Unique words | Total words | Avg. description length |
|---|---|---|---|
| Original | 1254 | 42928 | 18.06 |
| Syntax | 1212 | 38322 | 16.12 |
| Voice | 1216 | 39603 | 16.66 |
| Modifier | 1273 | 42011 | 17.67 |
| Accent | 1343 | 50533 | 21.26 |
| Tone | 1175 | 48726 | 20.50 |

datasets: ScanQA (Azuma et al., 2022) for 3D-VQA, NR3D (Achlioptas et al., 2020) and Scan-Refer (Chen et al., 2020) for 3D-VG. These rephrased sentences adhere to our variant definitions while retaining their original meaning. Formally, let $\mathcal{D} \in \{\text{ScanQA}, \text{NR3D}, \text{ScanRefer}\}$ and $o \in \{\text{Syntax}, \text{Voice}, \text{Modifier}, \text{Accent}, \text{Tone}\}$.

We utilize the LLM (*gpt-3.5-turbo*), for this sentence rephrasing task. Our methodology involves designing specific prompts, incorporating structured prompting and Chain of Thought (CoT (Wei et al., 2022)) techniques to guide paraphrasing. An abstract version of this prompt, as shown in Fig. 3, begins with a detailed and precise definition of variant derived from linguistic theories, represented as "style" in Fig. 3. This step aims to familiarize the LLM with the desired style. Following this, we provide three pairs of human-written, in-context examples as seeds to help the LLM understand the source style and achieve better rephrasing quality. The complete version of our prompt is available in the suppl.

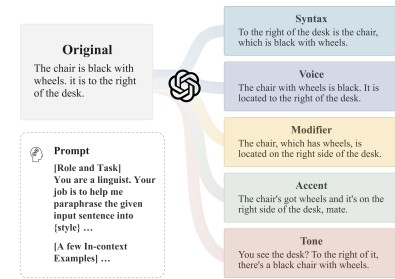

Figure 3: The rephrasing process to build our evaluation dataset.

To manage costs, we only utilize 25% of each dataset's data. For 3D-VG datasets ScanRefer and NR3D, we sub-sample from the official test split. And for ScanQA, we sample from the validation split. We employ uniform sampling for the NR3D subset from Referit3D (Achlioptas et al., 2020), which categorizes difficulty levels based on the number of object nouns in the natural language sentences and offers view-dependent and view-independent splits for more refined assessment. This ensures that our 25% subset maintains the integrity of the original data distribution.

**Basic Statistics and Quality Assessments.** We build the **3D L**anguage **R**obustness dataset, namely **3D-LR**, for our systematic benchmark. 3D-LR covers 2 tasks: *ScanRefer-R*, *NR3D-R* for 3D-VG and *ScanQA-R* for 3D-VQA. Within each task split, there are six different subsets, covering five language variants and one original version without rephrasing for comparison.

• *Basic Statistics.* Our *ScanRefer-R* has 2377 sentences, *NR3D-R* includes 1870 utterances, and *ScanQA-R* contains 1168 questions. Table 1(b) presents basic statistics of *ScanRefer-R*, comparing original sentences to their rephrased versions across different characteristics. Our primary focus is on simulating diverse grammatical expression structures rather than lexical richness. To this end, we implemented strict rules ensuring that paraphrasing does not alter the meaning or core object nouns. This approach resulted in similar unique words across the different dataset splits. In the accent and tone split, aiming to emulate a conversational style, we added some verbal phrases to the original sentences. This modification increased the average length of descriptions.

To further analyze the diversity characteristic of our dataset and its comparators, we visualized the syntax diversity of our dataset using vectorized syntax trees that represent sentence structures. (Fig. 2(b)). First, we extract the sentence structures into syntax trees. Then, we vectorize these trees and perform Principal Component Analysis (Pearson, 1901) (PCA) to project them onto a 2-dimensional space. Finally, we create a density map based on the features. Before creating the density map, we combined all five variant splits. For comparison, we analyzed a fully human-annotated NLP dataset, the Open Assistant (Köpf et al., 2023), which models human conversation (Fig. 2(d)). The density maps of both datasets exhibit similar spreading patterns and area sizes. This similarity suggests that our dataset accurately reflects the diversity of human language.

• *Dataset Quality Assessment.* To ensure the quality of rephrasing the dataset to facilitate a fair assessment, we further evaluate the rephrased dataset's quality, centered on preserving original sentence meanings. We employ traditional metrics and language model methods to check this. Specif-

Table 2: Experimental results on 3D-VG tasks with/without our pre-alignment module. The best results are in **bold**. ORACLE refers to the original dataset performance.

**(a)** Results of 3D-VG models on ScanRefer with predicted proposal. Measured in accuracy@kIoU

| Backbone | Method | Syntax | | Voice | | Modifier | | Accent | | Tone | |
|---|---|---|---|---|---|---|---|---|---|---|---|
| | | Acc@0.25 | Acc@0.5 | Acc@0.25 | Acc@0.5 | Acc@0.25 | Acc@0.5 | Acc@0.25 | Acc@0.5 | Acc@0.25 | Acc@0.5 |
| ORACLE: (Acc@0.25 = 42.36 Acc@0.5 = 27.68) | | | | | | | | | | | |
| ScanRefer (Chen et al., 2020) | baseline | 11.32 | 7.66 | 19.73 | 13.50 | 17.04 | 11.49 | 12.79 | 8.79 | 9.55 | 6.86 |
| | w. ours | **24.95** | **16.45** | **22.17** | **14.35** | **21.33** | **14.77** | **24.48** | **15.90** | **26.42** | **17.29** |
| ORACLE: (Acc@0.25 = 41.27 Acc@0.5 = 33.74) | | | | | | | | | | | |
| MVT (Huang et al., 2022) | baseline | 28.99 | 23.85 | 31.09 | 25.70 | 33.66 | 27.89 | 38.12 | 31.76 | 29.70 | 24.57 |
| | w. ours | **38.50** | **31.89** | **34.58** | **28.36** | **37.53** | **30.96** | **40.05** | **33.40** | **38.66** | **31.85** |
| ORACLE: (Acc@0.25 = 40.73) | | | | | | | | | | | |
| Chat-3D-v2 (Wang et al., 2023b) | baseline | 38.55 | - | 33.81 | - | 36.54 | - | 40.48 | - | 35.03 | - |
| | w. ours | **39.39** | - | **35.28** | - | **38.17** | - | **40.65** | - | **38.13** | - |

**(b)** Results of 3D-VG task with GT box proposal on NR3D. We measure listening accuracy of Referit3D and SAT.

| Backbone | Method | Syntax | Voice | Modifier | Accent | Tone |
|---|---|---|---|---|---|---|
| ORACLE: (Acc = 35.1) | | | | | | |
| Referit3D (Achlioptas et al., 2020) | baseline | 25.7 | 22.1 | 24.0 | 27.9 | 21.7 |
| | w. ours | **28.1** | **28.5** | **28.1** | **30.4** | **33.7** |
| ORACLE: (Acc = 48.8) | | | | | | |
| SAT (Yang et al., 2021b) | baseline | 43.9 | 34.2 | 38.9 | 41.2 | 34.9 |
| | w. ours | **45.1** | **44.1** | **41.6** | **44.6** | **46.9** |

**(c)** Results of 3D-VG task with GT proposal on ScanRefer. Measuring listening accuracy of Referit3D and SAT.

| Backbone | Method | Syntax | Voice | Modifier | Accent | Tone |
|---|---|---|---|---|---|---|
| ORACLE: (Acc = 34.8) | | | | | | |
| Referit3D (Achlioptas et al., 2020) | baseline | 16.8 | 23.9 | 24.0 | 24.0 | 17.9 |
| | w. ours | **31.9** | 23.2 | **24.5** | **27.2** | **34.4** |
| ORACLE: (Acc = 40.7) | | | | | | |
| SAT-NR3D (Yang et al., 2021b) | baseline | 37.1 | 33.2 | 36.6 | 34.7 | 33.1 |
| | w. ours | **39.7** | **35.7** | **36.7** | **38.2** | **41.0** |
| ORACLE: (Acc = 53.9) | | | | | | |
| SAT-ScanRefer (Yang et al., 2021b) | baseline | 36.7 | 40.6 | 43.7 | 50.3 | 37.0 |
| | w. ours | **51.6** | **44.5** | **47.6** | **52.5** | **50.7** |

ically, we measure the average edit distance (Ristad & Yianilos, 1998) between each original and rephrased sentence, with results presented in Table 1(a). A smaller edit distance value signifies a closer sentence structure, indicating minimal alteration. Additionally, we assess semantic integrity using cosine similarity between sentence vectors derived from a neural language model BERT (Devlin et al., 2019) and Glove (Pennington et al., 2014) of the original and rewritten sentences. Higher scores suggest a greater preservation of semantic content. More explanations of the quality assessment method are detailed in the Supplementary file. Our findings, underscored by manual review, reveal that the rephrased dataset largely maintains its semantic essence. Notably, the observed reduction in cosine similarity scores below 0.9, particularly in accent-based sentence rephrasing, can be attributed to adding some spoken expressions and phrases (such as "Hey bro"). After humans verify, the underlying meaning remains.

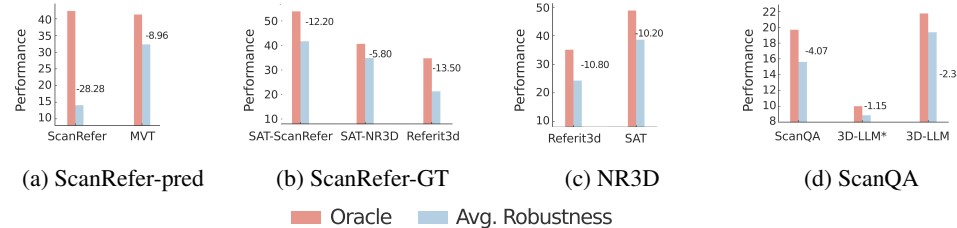

(a) ScanRefer-pred    (b) ScanRefer-GT    (c) NR3D    (d) ScanQA

■ Oracle    ■ Avg. Robustness

Figure 4: Performance summary of existing models on our 3D Language Robustness benchmark. Acc@kIoU for ScanRefer(a), ScanRefer-GT(b), Listening accuracy (Acc) for NR3D(c) and EM@1 for ScanQA(d) are measured. Average robustness is computed over all five language variant splits. It shows performance drops in 3D-QA and 3D-VG models, indicating a lack of robustness.

## 4 EXPERIMENTS

**Evaluation Method.** We benchmark different models of both 3D-VG and 3D-VQA. For 3D-VG with predicted proposals, we assess ScanRefer, MVT and Chat-3D (Wang et al., 2023b) using the proposed *ScanRefer-R*. For 3D-VG with Ground Truth (GT) proposals, we evaluate Referit3D and SAT (Yang et al., 2021b) on our *NR3D-R*. In 3D-VQA, *ScanQA-R* serves as the benchmark, including evaluations of ScanQA (Azuma et al., 2022), 3D-LLM (Hong et al., 2023) LLaVA-3D (Zhu et al., 2024) and Video3D-LLM (Zheng et al., 2024). Except for SAT, all models are trained using their corresponding training sets. We explore both models for SAT from two training settings: training on original NR3D (Achlioptas et al., 2020) and training on ScanRefer.

**Metrics.** For the 3D-VG task with predicted proposals, we use Acc@kIoU (k set to 0.25 and 0.5) to evaluate the accuracy, considering IoU thresholds of the box prediction. In 3D-VG tasks with ground truth proposals, we employ listening accuracy (Acc), where a model scores 1 for correctly selecting the target object from a list of candidates and 0, otherwise, as defined in Referit3D (Achlioptas et al., 2020). For 3D-VQA, following ScanQA (Azuma et al., 2022), we report Exact Match@k (EM@k) for top-k prediction accuracy. Due to page limitations, other metrics, e.g. BLEU-1 (Papineni et al., 2002), CIDEr (Vedantam et al., 2015) and their explanations, are provided in the supplementary file.

## 4.1 MAIN RESULTS

We systematically evaluate various models for 3D-VG and 3D-VQA. The results are summarized in Fig. 4. Oracle exhibits its performance on the unaltered test set, and we calculate average robustness by averaging its performance across all five language variant splits. It measures the overall performance degradation when the model encounters unfamiliar linguistic patterns (absent from the training set). We noted a serious performance drop in all models when confronted with language variants, indicating a lack of robustness in existing 3D-VL models.

**3D Visual Grounding.** Table 2(a) - Table 2(c) present results for 3D Visual Grounding with Ground Truth (GT) and predicted bounding boxes on different datasets. Referit3D and SAT models (trained on different datasets) show significant performance decreases in tone and syntax splits, with Referit3D dropping by up to 18% in simple syntax splits, (Table 2(c)), indicating brittleness and overfitting to sentence patterns. This trend is consistent across datasets, including NR3D (Achlioptas et al., 2020), (Table 2(b)), suggesting a bias towards the implicit pattern in these datasets. Examining models with integrated point cloud detectors, like ScanRefer, MVT (Huang et al., 2022), we observe a similar pattern of sensitivity to language style changes, shown in Table 2(a). ScanRefer shows a substantial performance decrease of 32.81% in Acc@0.25 on tone splits, while MVT, using the more robust *bert* text encoder, shows a smaller yet significant drop, particularly in syntax splits. We also evaluated Chat-3D-v2 at Acc@0.25 following the official setting and observed similar performance drop to language variations. These results highlight that existing 3D-VG models, regardless of whether they use detectors or not, are sensitive to variations in language styles.

**3D Visual Question Answering.** Table 3 outlines the 3D-VQA task results. Like in the 3D-VG task, the ScanQA model (Azuma et al., 2022) shows performance degradation across all paraphrased style splits. The LLM-based 3D models (3D-LLM (Hong et al., 2023), LLaVA-3D, and Video3D-LLM), despite being pre-trained on large datasets, also experience reductions in performance across most splits but demonstrate greater robustness than other models. This resilience is attributed to their language model backbones' extensive training on large corpora. Notably, in our modifier split, the un-fine-tuned 3D-LLM* variant surpasses the ORACLE in performance, and similar patterns are observed with LLaVA-3D. This is likely because LLM-based models more effectively handle common language expressions, as seen in the modifier of nouns. Among these models, Video3D-LLM shows the strongest overall performance. Further discussions are shown in the supplementary file. Interestingly, 3D-LLM's fine-tuning on ScanQA faces a severe performance drop on the Tone split, suggesting that fine-tuning may cause catastrophic forgetting and lead to a biased feature space.

## 5 ANALYSIS AND IMPROVED MODEL

The systematic evaluation results indicate a lack of robustness in existing 3D-VL models. We analyze the reason behind this issue through an in-depth investigation. (Sec. 5.1) Next, we developed a simple yet effective plug-and-play method. (Sec. 5.2) This approach can be applied to any pre-trained model, enhancing its robustness to diverse natural language characteristics without the need for re-training or additional data augmentation. Finally, we discuss data augmentation, a technique commonly used to enhance model performance and robustness. (Sec. 5.3)

Table 3: Evaluating 3D-VQA models with/without our pre-alignment module on ScanQA. 3D-LLM* denotes the model without task-specific fine-tuning. More metrics are in the Appendix.

| Method | Syntax | Voice | Modifier | Accent | Tone |
|---|---|---|---|---|---|
| ORACLE: (EM@1 = 19.69) | | | | | |
| ScanQA | 14.98 | 17.12 | 16.01 | 14.04 | 15.92 |
| w. ours | **19.26** | **18.58** | **18.32** | **18.41** | **18.49** |
| ORACLE: (EM@1 = 10.02) | | | | | |
| 3D-LLM* | 8.99 | 9.50 | 10.27 | 8.13 | 7.45 |
| w. ours | **10.19** | **9.59** | 9.93 | **9.25** | **9.25** |
| ORACLE: (EM@1 = 21.75) | | | | | |
| 3D-LLM | 21.06 | 19.61 | 19.86 | 19.52 | 16.78 |
| w. ours | **21.15** | 19.52 | **20.80** | **19.95** | **20.80** |
| ORACLE: (EM@1 = 27.40) | | | | | |
| LLaVA-3D | 27.05 | 24.91 | 26.54 | 23.89 | 22.26 |
| w. ours | **27.48** | **26.03** | **26.63** | **25.94** | **26.20** |
| ORACLE: (EM@1 = 32.95) | | | | | |
| Video3D-LLM | 30.46 | 28.79 | 29.53 | 27.31 | 20.41 |
| w. ours | **31.91** | **29.27** | 29.10 | **29.62** | **29.78** |

## 5.1 WHY DO THE 3D-VL MODELS FAIL?

Our systematic evaluation identifies that the lack of robustness is consistently and commonly observed across various models and tasks. Suggesting existing models fail at aligning 3D modality with text. We further conduct an in-depth analysis to figure out "Why do 3D-VL models fail?".

**The Fragility of Fusion Space.** We notice that in Table 2(a), though MVT is equipped with a pre-trained BERT (Devlin et al., 2019), it still faces performance dropping issues. Therefore, we hypothesize the real problem is the feature fusion module used to fuse point cloud features generated by the vision encoder (Qi et al., 2017a;b; 2019) and text features generated by the text encoder.

To study this, we use the syntax variants from our *ScanRefer-R* as an example. We measure the sentence vector cosine similarity between the original sentence and the syntax variants before the fusion module (right after the BERT) and after passing the fusion module. We analyze all failure cases in this split by calculating the similarity between sentence embedding features or object features of the syntax variants and their corresponding original sentences. Subsequently, we construct the probability density function (PDF), as illustrated in Fig. 5. The PDF for cosine similarity between text pairs reveals distinct patterns before and after the fusion process. For text features, distribution before fusion (red distribution in Fig. 5(a)) is notably skewed towards higher similarity values, suggesting that the original sentences and their syntax variations (not seen in the training set) are closely aligned. This indicates that the text encoder (e.g., BERT) is robust enough to handle variations well.

In contrast, the cosine similarity distribution of text features post-fusion (blue distribution in Fig. 5(a)) is more widespread, with a significant portion extending towards lower similarity values. This suggests that the fusion process negatively affects the integrity of text features. The robustness of the fusion module is weak towards sentence variations. It performs poorly when given syntax-altered sentences absent from the training data. In other words, the fusion module is biased towards the training dataset, rather than genuinely understanding the semantics of natural language.

Moreover, Fig. 5(b) depicts the distribution of object features, calculated analogously to the text similarity previously discussed. Given that the visual scene is identical for both syntax-split and original sentences, the expected similarity value should be exactly 1. Nonetheless, the green distribution in the figure shifts towards the left, indicating lower similarity values. This shift mirrors the trend observed in the text feature distribution, implying a negative impact on the object features as well. MVT uses a Transformer-based fusion module with attention mechanism, each object token can attend to all text tokens. Thus, the shift in object similarity likely stems from the amplification of subtle text differences through attention, highlighting the model's sensitivity to text variations. This further supports the fragility of the fusion feature space and a bias toward the training set, rather than true semantic understanding. Also, note that absolute similarity *has no physical meaning* in high-dimensional space, therefore we *focus on the relative comparison results*.

**Low Diversity of Existing Datasets.** We further identify that the major cause behind the above phenomenon is rooted in the low diversity of the original dataset and its large domain gap to the human daily language. It hinders the model from learning more general information. For example, we study the syntax diversity using the ScanRefer (Chen et al., 2020) dataset. Specifically, we propose to use a vectorized syntax tree to represent a sentence, then we conduct PCA analysis (Maćkiewicz & Ratajczak, 1993) and plot the density map. The darker region shows more sentences are being projected to this place. We can tell from the density map that the origin 3D-VL dataset ScanRefer has a simpler syntax pattern (the more compact darker region in Fig. 2(a)) while the other dataset shows more spreading high-density regions. It indicates that the existing dataset lacks enough diversity to reflect real-world language characteristics properly. Moreover, given such a dataset to facilitate model training, the model can easily fit the simple implicit pattern without obtaining a generic understanding of language features. It is worth mentioning that our proposed dataset (Fig. 2(b)) shares a similar pattern with the open assistant (Köpf et al., 2023) dataset (Fig. 2(d)), which is a dataset that has a closer domain gap with daily language and enough diversity, indicating the high quality of our dataset. We further prove that increasing the dataset's diversity can reduce the robustness problem to a certain extent, as detailed in the discussion of data augmentation in Sec. 5.3.

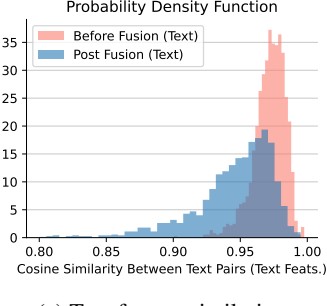
(a) Text feature similarity.

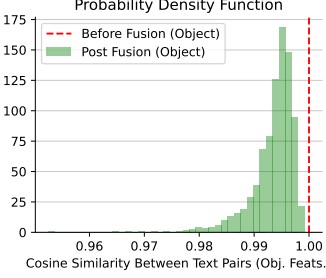
(b) Object feature similarity.

Figure 5: Probability Density Functions (PDFs) of cosine similarity between text pairs of original Sentences and their corresponding syntax variants.

## 5.2 PLUG AND PLAY PRE-ALIGNMENT MODULE

**Pre-Alignment Module.** We have identified that the fusion module is sensitive to the textual input and shows overfitting to the language style pattern of the training data. Therefore, we propose a training-free pre-alignment method to first convert the sentence pattern into the style that the trained model prefers. More specifically, we design an LLM-based parser, which maps sentences in any style into the models' preferred style. Since we have a well-trained model, we naturally assume that we can assess the training data. We propose to use LLM to conduct the style transfer, mapping sentences in any style into the format that the model is good at while maintaining the same meaning. We design a structured prompt containing "Generic Rules" and "In-Context Examples": the genetic rule contains format instructions for better post-processing, a simple sentence *"You should not change the meaning of the input sentence..."* to encourage the model to preserve most information. Since we can see what kind of data the model is trained from, we just need to give three to six in-context examples indicating the pattern we are dealing with and the style the model prefers. Empirical evidence suggests that for modern, powerful large language models, as few as 3 to 6 samples are sufficient to capture the desired style effectively. The pre-aligned sentence is directly fed into the same model for prediction. This process is called pre-alignment without the need for retraining. It also does not need large-scale additional annotation on different variants.

**Performance Benefit of Our Pre-Alignment Module.** When used in conjunction with existing models, our pre-alignment module achieves significant performance enhancements across various datasets, as demonstrated in Table 2 to Table 3. In particular, ScanRefer and MVT models exhibit substantial improvements across all language variants on *ScanRefer-R* with our module. For example, ScanRefer's Acc@0.25 increases by over 16% in the tone split. Similar improvements are observed in *NR3D-R* and 3D-VQA models. However, 3D-LLM (Hong et al., 2023) without fine-tuning does not benefit from our module because it effectively handles language expressions since it is a large-scale pre-trained language model. Table 4 further shows the performance of our proposed method with SAT (a grounding model). Our pre-alignment module recovered performance on all five splits compared with this baseline. Notably, our method can recover up to 12% accuracy on tone split without training. This indicates the effectiveness of our proposed module. More discussion is provided in the appendix.

## 5.3 DISCUSSION ON DATA AUGMENTATION

Here, we investigate whether data augmentation can address model robustness. We experimented with the NR3D (Achlioptas et al., 2020) dataset in two scenarios, with the result shown in Table 4. The first, "SAT w. aug - 40k", involved training on a mixed-style dataset, size-matched to the original NR3D, resulting in 40,000 samples balanced on five variants. The second, "SAT w. aug - 80k", doubled the dataset to 80,000 samples by merging training data variants. We discuss more details in the supplementary. Table 4 shows our method surpasses data-augmented models of equal size and rivals those trained on double the data. This highlights our method's effectiveness, especially considering the impracticality and high cost of obtaining exhaustive annotations for augmentation. This also demonstrates the significant demand for the augmentation of data.

Both data augmentation and our method hold potential.(i) We believe augmentation could improve performance, but it demands large and diverse high-quality datasets (Table 4). However, as human language exhibits over 23 styles, it is difficult to cover all for augmentation. (ii) Our method, requiring no training, offers immediate plug-and-play benefits to existing models and is ready for future boosts with more advanced LLMs. (iii) Our approach could offer diverse data for augmentation.

Table 4: Results of data augmentation.

| Backbone | Syntax | Voice | Modifier | Accent | Tone |
|---|---|---|---|---|---|
| SAT | 43.9 | 34.2 | 38.9 | 41.2 | 34.9 |
| SAT w. aug - 40k | 43.1 | 40.8 | 40.5 | 41.0 | 43.5 |
| SAT w. aug - 80k | **45.7** | 42.3 | **44.5** | 42.5 | 45.9 |
| SAT w.ours | 45.1 | **44.1** | 41.6 | **44.6** | **46.9** |

## 6 CONCLUSION

We present a novel task and benchmark dataset to study language robustness in 3D vision-language models. Our dataset, designed based on linguistic theories, includes diverse language variants. Through comprehensive evaluations, we identify significant fragility in existing 3D-VL models when encountering real-world language variations. We analyze the reasons and propose a simple yet effective solution without requiring additional training. This work aims to facilitate future research in improving model robustness, crucial for deploying 3D-VL models in real-world applications.

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

# A  APPENDIX

## A.1  THE USE OF LARGE LANGUAGE MODELS (LLMS)

Large language models (LLMs) were used in this work solely as a general-purpose assistance tool. Specifically, LLMs were employed to assist with grammar checking and polishing during the writing process. Additionally, in the construction of the dataset, LLMs were used to generate initial linguistic variations, which were subsequently reviewed, filtered, and refined by human annotators. LLMs did not contribute to the research ideation, technical design, or analytical conclusions of this work.

## A.2  MORE EXAMPLES

This section provides additional visual examples to further illustrate the language robustness challenge discussed in the main paper. Examples are presented in Fig. 11 to Fig. 15. We employ SAT (Yang et al., 2021b), a 3D Visual Grounding model trained on the ScanRefer dataset (Chen et al., 2020), to test its effectiveness in handling subtle linguistic variations. Specifically, we analyze its response to five sentence variants that are semantically identical to the original but phrased differently, reflecting natural variations in human communication. These examples underscore a key limitation: the model struggles with minor rephrasings, such as sentence inversions, that do not change the underlying meaning, highlighting a critical challenge in current natural language processing capabilities. Moreover, we visualize the predictions, shown in Fig. 9 to Fig. 10, made by plain SAT and SAT enhanced with our pre-alignment module to demonstrate the effectiveness of our approach.

## A.3  DATASETS

In this section, we first present basic statistics and semantic similarity analysis for two additional splits within our **3D Language Robustness** (**3D-LR**) dataset: *NR3D-R* for 3D Visual Grounding (3D-VG) and *ScanQA-R* for 3D Visual Question Answering (3D-VQA). We then delve into a detailed discussion about the variations in language usage within these datasets. Additionally, we outline the methodology employed in leveraging large language models (LLMs) to construct the **3D-LR** dataset. Lastly, we discuss the diversity of these datasets.

### A.3.1  BASIC STATISTICS

Table 5 detail the basic statistics of *ScanQA-R* and *NR3D-R* respectively. They compare original sentences with their rephrased counterparts, focusing on characteristics such as word count, noun usage, and sentence complexity. These statistics are akin to those observed for *ScanRefer-R*, as discussed in Section 3.2 of the main paper. This similarity indicates that our strict rephrasing rules effectively preserve the original meaning and core object nouns across different dataset splits. The consistent number of unique words across these splits corroborates this observation. For the modifier, accent, and tone variation, which aims to mimic conversational style, we incorporated additional verbal phrases and modifiers into the original sentences. This modification led to an increase in the unique words, total words, and average sentence length. Besides, we also use both neural semantic metrics and traditional metrics to ensure our dataset preserves the original meaning. The results are shown in Table 6. We can observe that all our paraphrased dataset has an edit distance to the original sentence lower than 1, which means they are only slightly different from the original sentence. Moreover, the high cosine similarity in BERT and Glove suggests that, from a neural network model's perspective, they almost mean the same.

### A.3.2  MORE DISCUSSION OF LANGUAGE VARIANTS

We provide several examples of sentence variants representing different aspects of human language as illustrated in the captions below the corresponding images from Fig. 11. These examples showcase five distinct groups of sentences derived from the original sentences in the ScanRefer dataset, highlighting variations in syntax, voice, modifier, accent, and tone.

Table 5: More basic statistics. We count the unique words, total words, and the average sentence length on our paraphrased dataset, named *ScanQA-R*, in (a) and *NR3D-R* in (b).

**(a)** Basic statistics of different data splits using *ScanQA-R* as an example.

|  | Unique words | Total words | Avg. description length |
|---|---|---|---|
| Original | 565 | 10257 | 8.78 |
| Syntax | 556 | 10731 | 9.19 |
| Voice | 566 | 10665 | 9.13 |
| Modifier | 611 | 11548 | 9.89 |
| Accent | 679 | 14085 | 12.06 |
| Tone | 606 | 14385 | 12.32 |

**(b)** Basic statistics of different data splits using *NR3D-R* as an example.

|  | Unique words | Total words | Avg. description length |
|---|---|---|---|
| Original | 909 | 19475 | 10.41 |
| Syntax | 907 | 20769 | 11.11 |
| Voice | 928 | 21785 | 11.65 |
| Modifier | 1047 | 22616 | 12.09 |
| Accent | 1002 | 22857 | 12.22 |
| Tone | 873 | 30702 | 16.42 |

Table 6: More semantic similarity statistics. We measure the the cosine similarity between our *ScanQA-R* (a) and *NR3D-R* (b) with the original data using BERT embedding, Glove embedding. We also compare the Edit Distance (ED). This suggests that our data effectively retains the original's meaning.

**(a)** Semantic similarity analysis of our *ScanQA-R*.

|  | BERT Sim. ($\uparrow$) | Glove Sim. ($\uparrow$) | ED ($\downarrow$) |
|---|---|---|---|
| Syntax | 0.94 | 0.98 | 0.40 |
| Voice | 0.96 | 0.98 | 0.72 |
| Modifier | 0.94 | 0.93 | 0.65 |
| Accent | 0.84 | 0.85 | 0.53 |
| Tone | 0.87 | 0.92 | 0.48 |

**(b)** Semantic similarity analysis of our *NR3D-R*.

|  | BERT Sim. ($\uparrow$) | Glove Sim. ($\uparrow$) | ED ($\downarrow$) |
|---|---|---|---|
| Syntax | 0.90 | 0.98 | 0.53 |
| Voice | 0.87 | 0.96 | 0.48 |
| Modifier | 0.86 | 0.90 | 0.45 |
| Accent | 0.89 | 0.93 | 0.64 |
| Tone | 0.81 | 0.85 | 0.41 |

Humans employ many language styles to convey their intentions, meanings, and emotions, and emphasize certain aspects (Barber et al., 2009). These language styles allow for subjective expressions and the inclusion of emotions, enriching the communication experience.

**Syntax**, for instance, plays a crucial role in human language by providing various word or phrase orders to create different sentence structures. Inverse sentences, such as posing a question before providing the answer or lifting the part that one wants to highlight to the start of the sentence, are common in daily conversations to engage the listener and evoke curiosity or anticipation.

**Voice**, on the other hand, enables individuals to transform sentences from active to passive voice or vice versa. This intentional use of voice enables slight changes in emphasis, highlighting different subjects or actions in a sentence, and can evoke various emotional responses from the listener.

**Modifiers**, such as adjectives and adverbs, are extensively used by humans to enhance the details and depth of their expressions. By incorporating a wide variety of modifiers, individuals can add subjective elements, convey emotions, and paint vivid pictures in the minds of their audience.

**Accent** reflects the distinct linguistic habits of English speakers from different regions, contributing to the diversity and richness of language. It allows individuals to express their cultural identity and provides a unique flavor to their communication, introducing subjective nuances and regional colloquialisms.

Lastly, **tone** encompasses the attitudes and emotions conveyed in a sentence. For example, by using questions, employing irony, or other techniques, individuals can add layers of meaning and emphasize certain aspects, evoking specific emotional responses from their audience.

### A.3.3 DETAILED PROMPT FOR DATASET CONSTRUCTIONS

We employ structured prompting and chain of thought (Wei et al., 2022) techniques to create language variants. Please note that the "style requirement" slot specifies the desired variant type in strict rules, and the "sentence" slot should be filled with the original sentence for rewriting. Our process ensures high-quality rewrites while adhering to these parameters. Fig. 6 shows the overall structure of our prompt.

### A.3.4 More Discussions on Dataset Diversity

Fig. 8 displays the syntax diversity of the original NR3D (Achlioptas et al., 2020), ScanQA (Azuma et al., 2022), our proposed 3D-LR dataset, and other datasets (Chen et al., 2015; Köpf et al., 2023) for comparisons, utilizing vectorized syntax trees. Specifically, we employ the NLTK (Bird et al., 2009) to construct syntax trees, which represent the syntactic structure of sentences. These trees are then transformed into strings. Subsequently, we apply the Term Frequency-Inverse Document Frequency (TF-IDF) (Salton & Buckley, 1988) technique, using TfidfVectorizer, to convert these syntax strings into quantifiable features. Following this, we implement Principal Component Analysis (PCA) (Pearson, 1901) to project these features into a lower-dimensional space, enabling us to plot the density map for each dataset. In these maps, smaller dark areas signify more uniform or similar sentence patterns, indicating a lack of diversity in sentence structures within the dataset. Comparing Fig. 8(d) to Fig. 8(f), which represents a large-scale, fully human-annotated dataset, namely Open Assistant (Köpf et al., 2023), we see a similar pattern. This similarity suggests that our proposed *NR3D-R* dataset accurately reflects the characteristics of natural language. On ScanQA (Azuma et al., 2022), the original dataset shows a strong, compact pattern (Fig. 8(b)), which model is easy to utilize such shortcut, while our *ScanQA-R* (Fig. 8(e)) improves the diversity and makes the pattern close to natural language in real-world communication.

### A.3.5 Human Evaluation and Annotator Agreement

To ensure the quality and reliability of our dataset, we conducted systematic human evaluations following a two-stage process. Initially, two annotators performed manual reviews on uniformly sampled subsets of the data (200 samples per split) to verify correctness and consistency. Given the large scale of the dataset (over 10,000 samples) and limited annotator resources, this review process was supplemented with rule-based and large language model (LLM) based checks applied to the entire dataset.

We further performed an additional human evaluation to assess inter-annotator agreement specifically on semantic equivalence and naturalness of the rephrased sentences. Three annotators were instructed to perform binary evaluations on: (a) whether the semantic meaning remained consistent between original and rewritten sentences, and (b) the naturalness of the rewritten sentences.

This evaluation was conducted on a uniformly sampled subset of 300 data points (balanced across all three splits) due to labour constraints. Inter-annotator agreement was measured using Fleiss' Kappa, yielding the following results:

- **Semantic equivalence agreement:** 0.561 (moderate agreement)
- **Naturalness agreement:** 0.708 (substantial agreement)

According to standard interpretation guidelines (where 0.41–0.60 indicates moderate agreement and 0.61–0.80 indicates substantial agreement), these scores demonstrate reasonable reliability for our benchmark. These results will be incorporated into the final version of the paper.

### A.4 Detailed Method Explanations

**Pipeline of Our Pre-Alignment Module** We introduce a method that leverages a large language model to enhance the performance of existing trained models on various language variants without necessitating re-training. This model aligns the sentence style with the training data of the original model. Specifically, it normalizes the input sentence to match the style of the data on which the models were initially trained. Fig. 7 outlines the overall pipeline.

### A.5 More Metrics of Systematic Evaluation

**Bilingual Evaluation Understudy (BLEU Score)** The BLEU score (Bilingual Evaluation Understudy) was initially proposed to assess the quality of machine translation systems (Papineni et al., 2002). It quantifies the similarity between a language model's translated output and the reference ground truth sentence. In the context of 3D Visual Question Answering, where models generate words, phrases, or short sentences as responses, the BLEU score evaluates the closeness of these model predictions to the correct answers. This metric effectively measures the linguistic accuracy

```
[Role and Task]
You are a linguist.  Your job is to assist
me in paraphrasing the input query according
to my needs while preserving its meaning
and critical elements.  [Style Requirement:
e.g., mimic the accent of an English Speaker
from a different region.]]

[Rules]

    1. Convert the given sentence into a more
       relaxed, conversational tone.

    2. Maintain the original meaning without
       altering it.

    3. Retain essential elements such as
       objects, attributes, relationships,
       and keywords.

    4. Present the revised sentence in JSON
       format, using the key "new_sentence"
       for the output.
[Example]
#example 1
sentence:  the dark blue pillow on the
papasan chair
return answer:  {{
"new_sentence":  "The dark blue pillow
resting upon the Papasan chair."
}}
#example 2 ...
#example 3 ...

[Detail Format Instruction]
You should ONLY return the JSON dictionary.
Python must be able to parse the response
into JSON.

#Begin Task

The sentence:  <{sentence}>
```

Figure 6: Overview of our prompt. To avoid complexity, we provide an example showcasing the overall structure of our prompt designed for generating sentence variants. All prompts we used, the dataset generation code, and the pre-generated dataset will be released later.

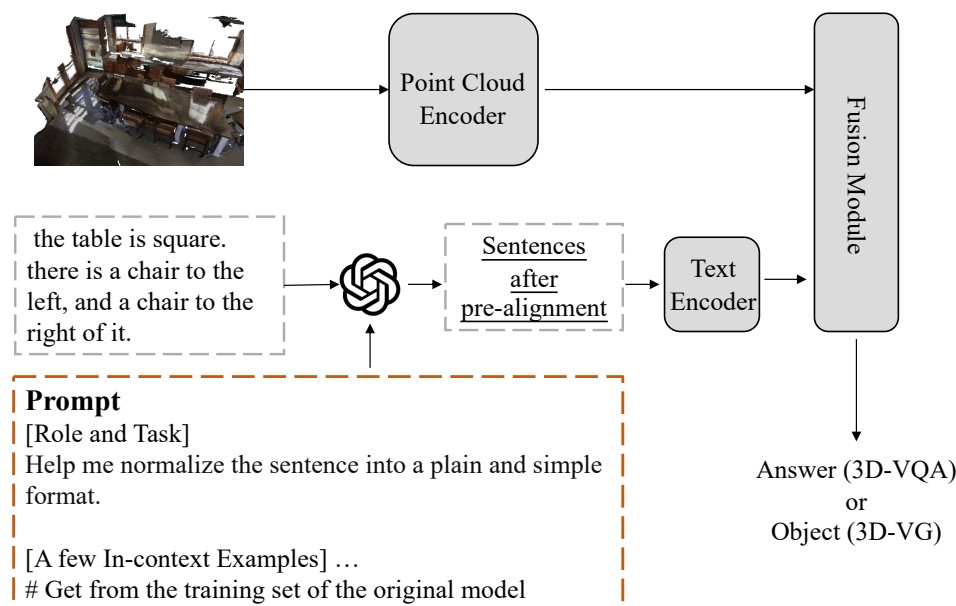

Figure 7: Overview of inference pipeline with our pre-alignment module. The entire pipeline is frozen without any training.

of answers in VQA tasks, especially for 3D-LLM (Hong et al., 2023), which generates the answer token by token rather than selecting from a candidate set.

BLEU-1 is a specific variant of the BLEU score that evaluates the precision of unigrams (individual words) in machine-generated text. BLEU-1 thus quantifies how many unigrams in the generated text accurately match those in the reference text, providing a metric for lexical accuracy in tasks like machine translation or automated content generation. A higher BLEU-1 score means that the answer is more accurate. For more details, please refer to (Papineni et al., 2002).

**CIDEr Score** The CIDEr (Vedantam et al., 2015) score originally evaluates the quality of image captions generated by computers. It calculates this score by comparing a generated sentence to a set of reference ground truth (GT) sentences. The key aspect of this comparison involves assessing the overlap of words and phrases between the generated caption and the GTs. This assessment is refined by weighting the n-grams (a contiguous sequence of $n$ items from a given sample of text) using TF-IDF, a technique that evaluates how frequently a word appears in the document relative to its commonness across all documents. The scoring favors captions that match the GTs closely and use terms specific to the given image context rather than generic words applicable to various images. Thus, in 3D-VQA, a higher CIDEr score indicates that the answer is accurate and contextually relevant.

### A.6 MORE EXPERIMENTAL RESULTS

We present additional results, including the BLEU-1 score and CIDEr score for the 3D-VQA task, in Table 7. These metrics further elucidate the performance of 3D-LLM (Hong et al., 2023), LLaVA-3D, and Video-3D-LLM, generative models that produce answers token by token.

### A.6.1 MORE DISCUSSIONS ON LLM-BASED MODELS

3D-LLM (Hong et al., 2023) is a recently proposed large scale 3D pre-training Vision-Language Model. The key idea is to use various 3D encoders to build unified 3D features. These features are first mapped into the same feature spaces as 2D images and then further used to train 2D VLMs. These 2D-VL models contain a powerful language model as a text encoder, e.g., Flan-T5 (Chung et al., 2022). Therefore, they have good language understanding abilities. Unlike ScanQA (Azuma et al., 2022) baseline for 3D-VQA, 3D-LLM and other LLM-based models (LLaVA-3D and Video-

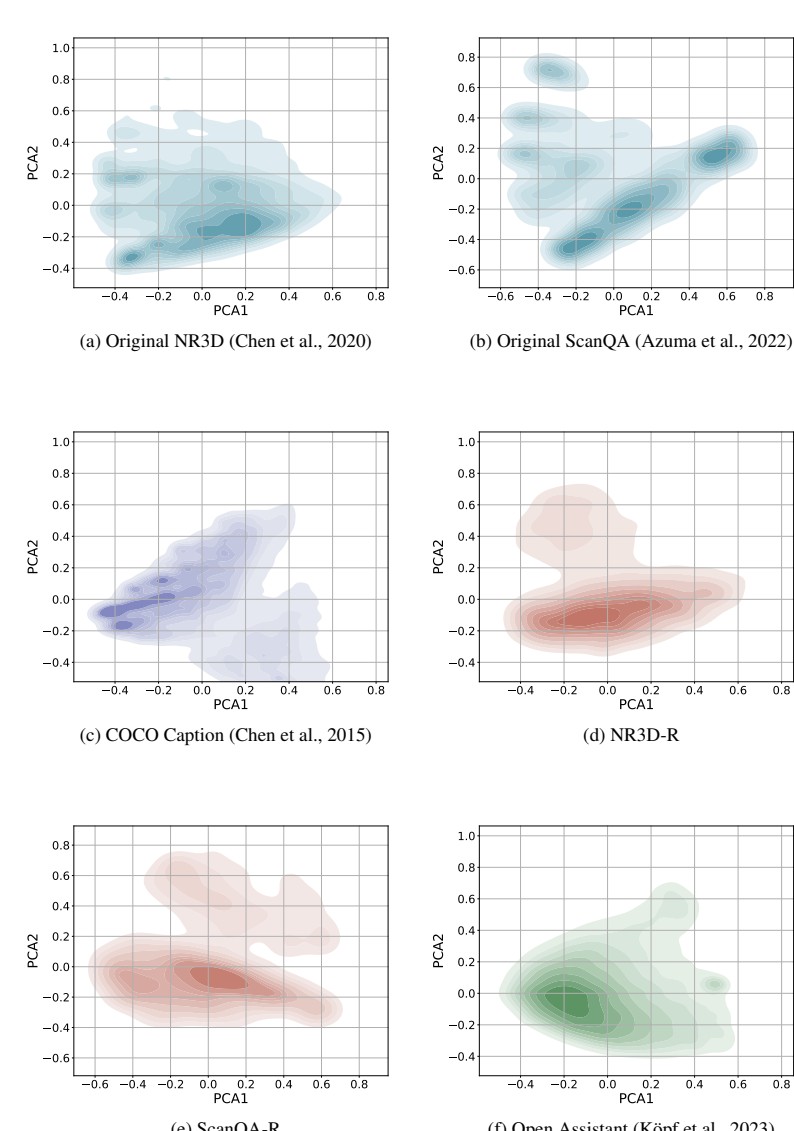

Figure 8: Density map of datasets' vectorized syntax structure principal features. Comparing *NR3D-R* and *ScanQA-R* with other representative datasets.

Table 7: Evaluating ScanQA, 3D-LLM, LLaVA-3D and Video-3D-LLM (pre-trained 3D vision language models), with/without our pre-alignment module on ScanQA, a 3D-VQA task. 3D-LLM* indicates the model before fine-tuning (FT) on ScanQA, while 3D-LLM shows post-FT results.

| Method | Syntax | | Voice | | Modifier | | Accent | | Tone | |
|---|---|---|---|---|---|---|---|---|---|---|
| | BLEU-1 | CIDEr | BLEU-1 | CIDEr | BLEU-1 | CIDEr | BLEU-1 | CIDEr | BLEU-1 | CIDEr |
| ORACLE: (BLEU-1 = 29.62 CIDEr = 60.42) | | | | | | | | | | |
| ScanQA | 21.18 | 44.16 | 26.98 | 53.06 | 26.55 | 52.89 | 22.51 | 44.16 | 22.83 | 48.21 |
| w. ours | **29.52** | **60.13** | **28.79** | **56.93** | **28.32** | **56.62** | **26.89** | **55.53** | **27.50** | **57.07** |
| ORACLE: (BLEU-1 = 23.13 CIDEr = 37.37) | | | | | | | | | | |
| 3D-LLM* | 20.19 | 33.55 | 21.97 | 33.85 | 22.41 | 38.03 | 15.45 | 32.03 | 18.52 | 28.05 |
| w. ours* | **23.42** | **37.48** | 20.90 | 33.63 | **22.57** | 36.18 | **21.37** | **33.44** | **21.82** | **34.55** |
| ORACLE: (BLEU-1 = 37.22 CIDEr = 74.00) | | | | | | | | | | |
| 3D-LLM | 38.20 | 74.21 | 36.52 | 70.79 | 36.06 | 70.00 | 35.05 | 68.28 | 22.73 | 50.85 |
| w. ours | 37.30 | 72.94 | 36.07 | 69.94 | **36.85** | **72.00** | **35.71** | 68.15 | **36.74** | **71.65** |
| ORACLE: (BLEU-1 = 44.16 CIDEr = 93.27) | | | | | | | | | | |
| LLaVA-3D | 41.43 | 88.57 | 42.07 | 87.19 | 42.72 | 90.34 | 31.88 | 73.12 | 27.64 | 67.43 |
| w. ours | **43.18** | **91.72** | **42.11** | **87.41** | 42.71 | **91.20** | **41.92** | **87.35** | **43.09** | **89.45** |
| ORACLE: (BLEU-1 = 47.80 CIDEr = 109.58) | | | | | | | | | | |
| Video-3D-LLM | 47.82 | 102.54 | 45.85 | 98.59 | 46.54 | 101.69 | 41.28 | 90.95 | 28.00 | 65.66 |
| w. ours | **48.25** | **105.77** | **46.22** | **99.97** | 45.98 | 100.18 | **45.11** | **98.86** | **46.79** | **102.15** |

3D-LLM) generate answers token by token. Therefore, we show the generative metrics in Table 7 to better assess their performance.

In experimental results, ScanQA exhibited significant performance degradation across all language variants, as evidenced by BLEU-1 and CIDEr metrics. Evaluating 3D-LLM in two configurations, solely pre-trained and fine-tuned on the ScanQA task, revealed distinct outcomes. "3D-LLM*" indicates the model before fine-tuning (FT) on ScanQA, while "3D-LLM" shows post-FT results. "3D-LLM*", without task-specific fine-tuning, shows a performance decline in almost all variant splits except for the modifier variant. In contrast, the fine-tuned "3D-LLM" demonstrates greater robustness compared to the non-VLM-based ScanQA model. Notably, variations in sentence modifiers had little impact on 3D-LLM across both training settings, suggesting that its extensive pre-training may equip it to handle such language variations. However, the fine-tuned "3D-LLM" experienced similar performance declines across most splits. Both pre-trained and fine-tuned models showed degradation in the tone split, which simulates stylistic diversity in daily communication. The fine-tuned "3D-LLM" exhibited more pronounced losses in the tone split than its solely pre-trained counterpart, indicating that while task-specific fine-tuning enhances downstream task performance, it may lead to catastrophic forgetting, reducing overall model robustness.

We also evaluated LLaVA-3D and Video-3D-LLM, two other LLM-based models for 3D-VQA. These models show similar patterns of sensitivity to language variations, particularly in tone and accent splits. The performance degradation observed across all these models highlights the general challenge of language robustness in 3D vision-language tasks, regardless of the specific architecture or pre-training approach.

### A.6.2 Ablation Results of different fusion modules

We conducted additional experiments on Referit3D, a model employing a GNN-based fusion module. Moreover, to further study the impact of different fusion architectures, we modified Referit3D's fusion module into two versions: a) vision language feature concatenation (no cross-modal interaction), and b) vision language feature average pooling. As shown in the Table below, all fusion variants exhibit performance degradation compared to ORACLE, with marginal differences between methods. This indicates the fragility persists regardless of architectural choices.

Table 8: Results of different fusion modules

| Backbone | ORACLE (Acc) | Syntax | Voice | Modifier | Accent | Tone |
|----------|--------------|--------|-------|----------|--------|------|
| Referit3D | 34.8 | 16.8 | 23.9 | 24.0 | 24.0 | 17.9 |
| Referit3D-avg | 34.5 | 15.8 | 24.2 | 23.5 | 21.5 | 20.0 |
| Referit3D-cat | 34.6 | 15.8 | 24.9 | 24.1 | 23.2 | 21.0 |

### A.6.3 Extended Language Style Evaluation

To further evaluate the robustness of our proposed method against diverse linguistic variations, we conducted additional experiments examining three scenarios: (a) repetition style (where speakers repeat key information when thinking), (b) metaphor style, and (c) mixed-style inputs. All experiments were conducted on a subsampled set of 250 data points to manage computational requirements while maintaining statistical significance.

The repetition style experiment was conducted on the same three models evaluated in the main paper, while the metaphor experiment utilized the Video-LLM-3D architecture to demonstrate our module's applicability to modern LLM-based solutions. For mixed-style inputs, we created a combined dataset incorporating all five linguistic styles from our main study (accent, tone, syntax, voice, and modifier), exposing models to randomized stylistic variations.

Table 9: Performance on Repetition Style Variants

| Method | Referit3D | SAT-NR3D | SAT-ScanRefer |
|--------|-----------|----------|---------------|
| Baseline | 26.6 | 44.7 | 50.6 |
| Ours | 34.6 | 45.2 | 53.8 |

Table 10: Performance on Metaphor Style Variants (Video-LLM-3D)

| Method | Metaphor |
|---|---|
| Baseline | 47.9 |
| w/ ours | 48.5 |

Table 11: Performance on Mixed Styles (Combined Dataset)

| Method | Referit3D | SAT-NR3D | SAT-ScanRefer |
|---|---|---|---|
| Baseline | 17.0 | 33.4 | 39.0 |
| Ours | 28.8 | 39.1 | 44.1 |

These results demonstrate our method's consistent effectiveness across diverse linguistic styles, with notable improvements over baseline approaches. The performance gains in mixed-style settings particularly highlight our module's generalization capabilities.

### A.6.4 MULTI-STYLE COMBINATION WITHIN SINGLE SENTENCES

We further investigated model performance when multiple stylistic variations are applied within individual sentences. This experiment applied 2-3 combined styles per sentence (e.g., metaphor + passive voice, syntax variation + tone, or modifier adjustments combined with other styles), with 124 sentences containing 2 styles and 126 sentences containing 3 styles.

Table 12: Performance on Multi-Style Sentences

| Method | Referit3D | SAT-NR3D | SAT-ScanRefer |
|---|---|---|---|
| Baseline | 17.0 | 41.3 | 49.8 |
| Ours | 31.4 | 44.6 | 53.0 |

The results align with trends observed in our main experiments: (a) baseline models struggle with complex multi-style inputs; (b) our method maintains robust performance; and (c) the module effectively handles within-sentence style mixtures, validating its generalizability across diverse linguistic challenges.

### A.6.5 MORE DISCUSSIONS ON OUR MODULE

Our pre-alignment module enhances the adaptability of trained models to different language styles without requiring re-training or data augmentations, as detailed in Table 7. Incorporating this module into the ScanQA baseline model resulted in a performance increase of approximately 8 BLEU-1 points and 16 CIDEr points. Furthermore, with our pre-alignment, ScanQA achieved performance on par with the ORACLE across all splits. For "3D-LLM*", not fine-tuned on the VQA task, our method provided additional robustness, particularly noticeable in the challenging Tone split. This trend was also evident in the fine-tuned (FT) "3D-LLM", where our pre-alignment module effectively compensated for the loss of diversity handling capabilities after fine-tuning.

### A.6.6 DATA AUGMENTATION

Section 5.3 and Tab. 4 of the main paper show that simple data augmentation, achieved by providing additional data to cover language variants for pre-training the model, still leads to unsatisfactory results. Despite the fact that a model trained on a more diverse dataset exhibits greater robustness to different language styles, its performance is worse than our straightforward, non-training-based method. Remarkably, our approach is comparable to, or even surpasses, the model trained on a dataset doubled in size, from 40,000 to 80,000 training samples. The underlying reason for this phenomenon is that while adding various styles into the training set gives the model an opportunity to learn different language variants, it simultaneously increases the learning difficulty. Consequently, the model fails to identify and overfit simple patterns within the dataset, as illustrated in Fig. 8. This also supports our hypothesis that existing models tend to learn shortcuts rather than achieving an actual understanding of natural language.

### A.7 LIMITATION AND BROADER IMPACT

This study's limitations arise from the potential incompleteness of the 3D-LR dataset in capturing the full spectrum of natural language variations. Natural language use by humans in daily communication is complex and varies widely. Despite this challenge, we have managed to summarize five major language characteristics from linguistic theories. These were used to build a language robustness dataset, which facilitates the systematic evaluation of existing 3D-VL models and identifies their vulnerabilities. While we provided an in-depth analysis to understand the reasons behind these phenomena, further studies need to focus on dataset quality, model training schemes, and the underlying causes of data augmentation failures.

This research impacts various domains, including embodied agents, autonomous navigation, robotics, and interaction with environments through language. By systematically studying and enhancing language robustness in 3D vision-language models, we offer practical benefits for applications that require the understanding of human instructions in 3D environments. The introduction of the 3D Language Robustness dataset (3D-LR) and the pre-alignment module not only demonstrates the potential for improving language robustness in 3D models but also sets a foundation for further exploration in this field.

### A.8 FUTURE WORKS

Future directions for this research include expanding robustness studies to other vision-language tasks and domains, such as 2D-VL, in contexts with limited data availability. A key area of investigation is understanding the low data efficiency of current data augmentation methods and discovering more efficient augmentation techniques. Another critical avenue is designing architectures that do not overfit dataset statistics and truly understand language. This research sets the stage for a more inclusive and comprehensive understanding of 3D vision-language models.

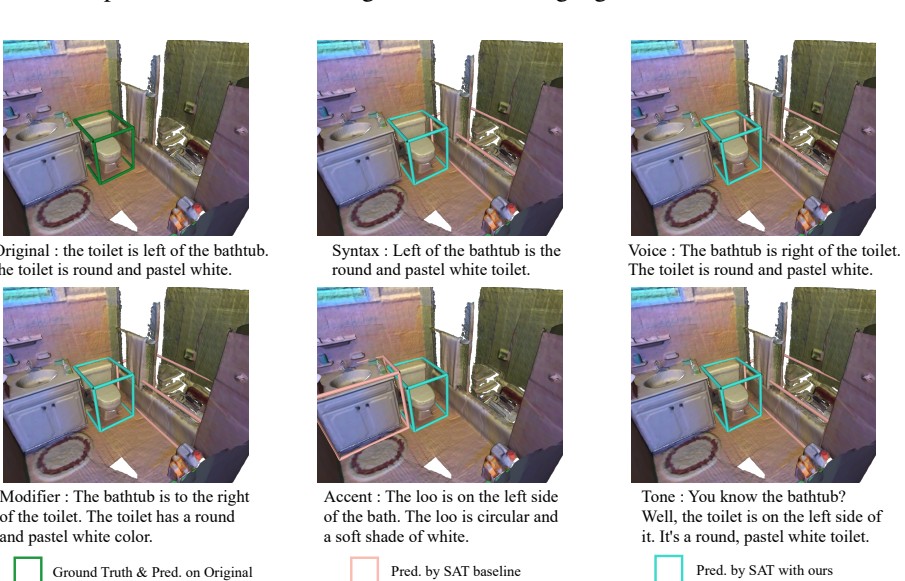

Figure 9: Predictions by SAT (Yang et al., 2021b) (a 3D Visual Grounding Model) on ScanRefer's sentence and five variants, compared with "SAT with Ours" using our pre-alignment module. Ground truth prediction is green-highlighted, and predictions from plain SAT are pink-highlighted. The figure shows SAT's difficulty with language variants, while our plug-and-play module aids in accurate predictions.

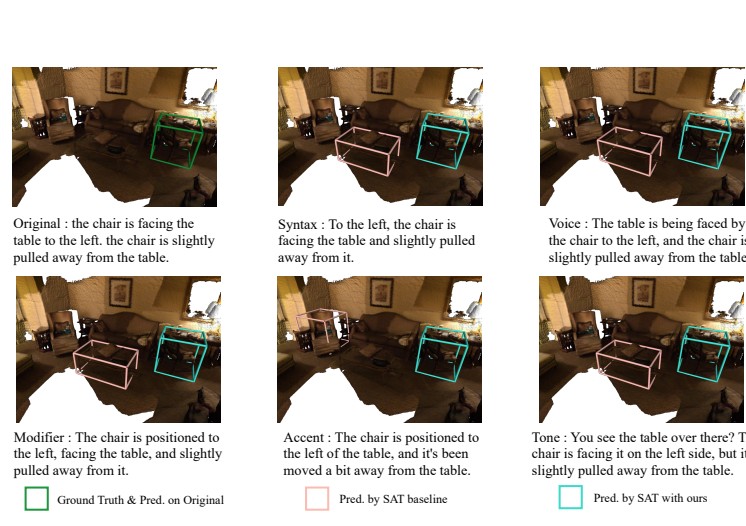

Figure 10: Predictions by SAT (Yang et al., 2021b) (a 3D Visual Grounding Model) on ScanRefer's sentence and five variants, compared with "SAT with Ours" using our pre-alignment module. Ground truth prediction is green-highlighted, and predictions from plain SAT are pink-highlighted. The figure shows SAT's difficulty with language variants, while our plug-and-play module aids in accurate predictions.

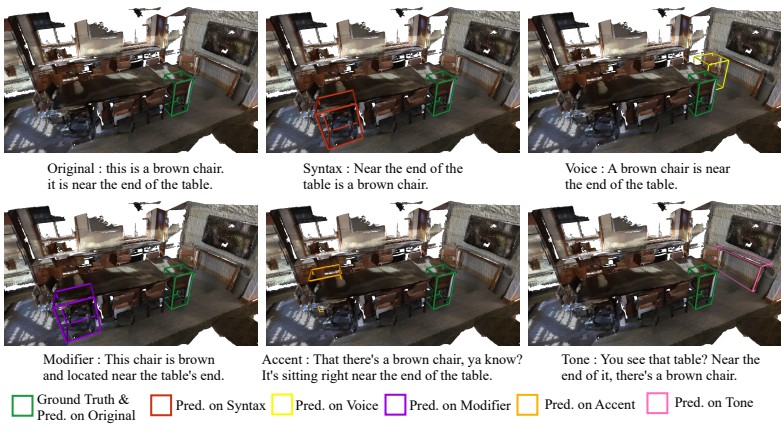

Figure 11: Prediction made by SAT (Yang et al., 2021b) (a 3D Visual Grounding Model) on Scan-Refer's original sentence and five modified variants. The ground truth and the SAT's prediction for the original sentence are highlighted in green. Differently colored boxes indicate the SAT's incorrect predictions on the sentence variants.

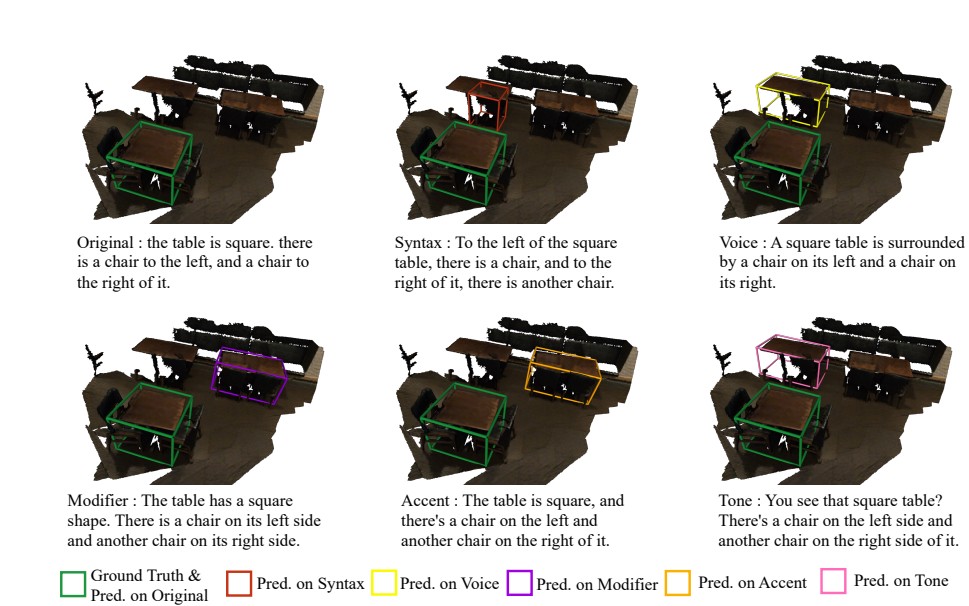

Figure 12: Prediction made by SAT (Yang et al., 2021b) (a 3D Visual Grounding Model) on Scan-Refer's original sentence and five modified variants. The ground truth and the SAT's prediction for the original sentence are highlighted in green. Differently colored boxes indicate the SAT's incorrect predictions on the sentence variants.

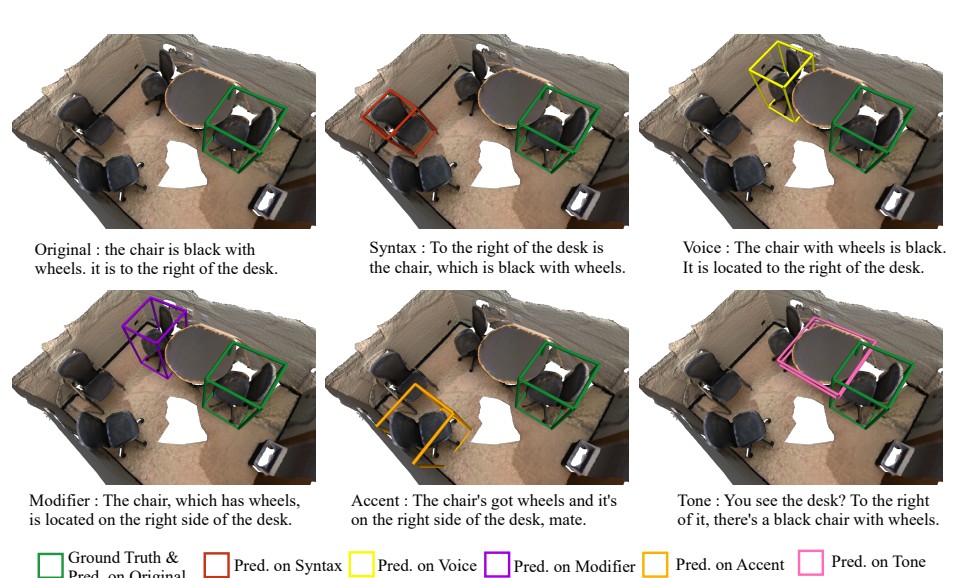

Figure 13: Prediction made by SAT (Yang et al., 2021b) (a 3D Visual Grounding Model) on Scan-Refer's original sentence and five modified variants. The ground truth and the SAT's prediction for the original sentence are highlighted in green. Differently colored boxes indicate the SAT's incorrect predictions on the sentence variants.

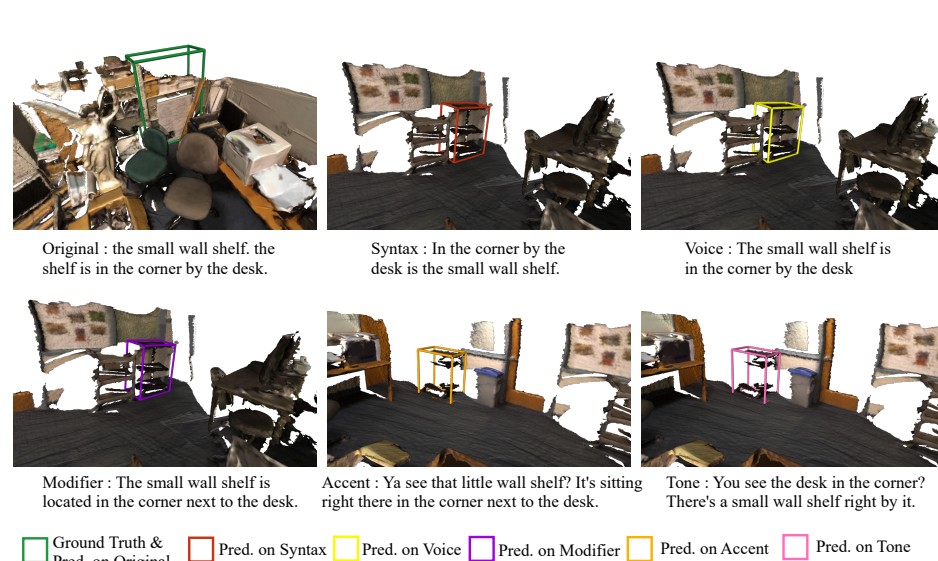

Figure 14: Prediction made by SAT (Yang et al., 2021b) (a 3D Visual Grounding Model) on Scan-Refer's original sentence and five modified variants. The ground truth and the SAT's prediction for the original sentence are highlighted in green. Differently colored boxes indicate the SAT's incorrect predictions on the sentence variants.

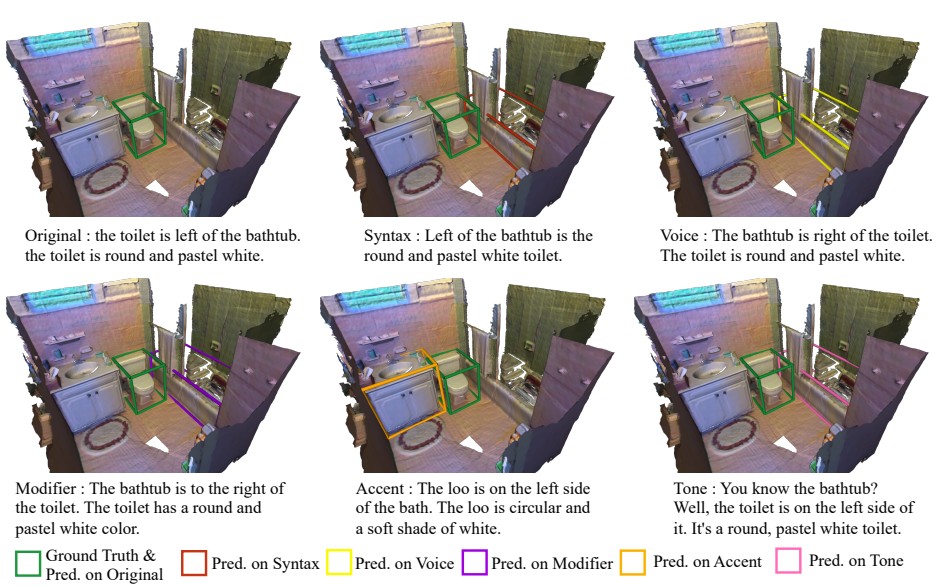

Figure 15: Prediction made by SAT (Yang et al., 2021b) (a 3D Visual Grounding Model) on Scan-Refer's original sentence and five modified variants. The ground truth and the SAT's prediction for the original sentence are highlighted in green. Differently colored boxes indicate the SAT's incorrect predictions on the sentence variants.

