# OpenReview forum: "Can 3D Vision-Language Models Truly Understand Natural Language?"
_ICLR.cc/2026/Conference — ICLR 2026 Conference Withdrawn Submission_

### Official Review · Reviewer_MUQe · 2025-10-22

**Soundness:** 1
**Presentation:** 2
**Contribution:** 2
**Rating:** 4
**Confidence:** 4

**Summary:**

The paper explores the sensitivity of the queried language for 3D MLLM. It finds that the fragility of fusion space and the low diversity of existing datasets like ScanRefer are two main reasons. Based on that, the paper proposes 3D Language Robustness (3D-LR) Benchmark, which is conducted on existing dataset ScanQA and NR3D, to evaluate the robustness of a 3D MLLM. In addition, it introduces a training-free method to transfer questions to the training style to improve the robustness for any 3D MLLM.

**Strengths:**

- It proposes a novel benchmark for the evaluation of the sensitivity of language for 3D MLLM.
- The plug and play pre-aligned module improves the robustness in the original MLLM.
- The writing is clear and easy to understand.

**Weaknesses:**

- Method. The plug and play pre-aligned module is simple and relies on the training data. It's hard to capture and filter helpful training data, which is unknown in most scenarios.
- The evaluated dataset is out-of-time. There are more advanced training data for 3D MLLM like PointLLM [1], JM3D-LLM [2] and so on. The paper should discuss more advanced methods and datasets.
- The improvements from the training-free module are more like fine-tuning. So the contribution is limited on the specific datasets, and it's unclear that if the more various dataset is the solution.

[1] PointLLM: Empowering Large Language Models to Understand Point Clouds, eccv 2024
[2] JM3D & JM3D-LLM: Elevating 3D Understanding with Joint Multi-modal Cues, tpami 2025

**Questions:**

- Why the Fig. 2(a) shows less density compared to the other data? The figures are different with Fig.8.
- I'm curious about the argumentation of voice. As for the voice, the difference is active or passive, which influences the order of tokens. It seems like the variable is position embedding. Is there any theory analysis about the effects?

---

### Official Review · Reviewer_svkm · 2025-10-30

**Soundness:** 3
**Presentation:** 3
**Contribution:** 3
**Rating:** 6
**Confidence:** 3

**Summary:**

This paper discovers an interesting problem with current 3D-VL models that they cannot understand variants of sentences. The author then constructs a dataset 3D-LR that covers 3DQA and 3DVG for comprehensive evaluation of this phenomenon. Analysis on the constructed benchmark showcase the lack of robustness of the current model w.r.t the language diversity. The paper also proposes to use LLM's rephrasing capability to mitigate this issue for 3D-VL models.

**Strengths:**

1. The paper is well-written and easy to follow and the problem it studies is interesting.

2. To evaluate the language understanding problem, the paper constructs a systematic approach for benchmarking it by proposing a dataset. The proposed dataset is verified and reliable.

3. The paper also identifies the robustness problem by grounding it to a specific module inside the model, namely the fusion module.

**Weaknesses:**

1. The proposed method for mitigating this problem serves as a good starting baseline but lack novelty and is not practically feasible.

2. The experiments are mostly done in domain-specific 3D-VL model. Despite the study of two LLM-based 3D-VL models, the paper overlooks the trend where generalizable LLM are capable of doing the 3D-VL tasks. I'm wondering whether other LLM-based 3D-VL models with better LLM backbone still suffers from the problem.

**Questions:**

N/A

---

### Official Review · Reviewer_nW8C · 2025-10-31

**Soundness:** 3
**Presentation:** 2
**Contribution:** 3
**Rating:** 4
**Confidence:** 4

**Summary:**

This paper investigates whether current 3D vision-language (3D-VL) models truly “understand” natural language.
The authors observe that these models perform poorly when encountering minor paraphrastic variations — such as changes in syntax, tone, or voice — even when the semantic meaning is identical.
To quantify this, they propose: (1) A dataset of linguistic variations derived from existing 3D-VL datasets (ScanRefer, NR3D, and ScanQA).
Variants are generated via LLM-based paraphrasing along five axes: syntax, voice, modifier, accent, and tone. This yields test splits where sentences retain identical semantics but differ in style. (2) They show that state-of-the-art 3D-VL models (e.g., SAT, MVT, ScanQA, 3D-LLM, LLaVA-3D, Video3D-LLM) suffer sharp performance drops (up to 32%) under these language variations. (3) A training-free LLM-based “style normalization” component that rewrites incoming user sentences into the model’s training-style syntax before inference. This boosts robustness by ~8–12% on average — matching models trained with twice as much augmented data.

The paper claims that (a) current 3D-VL models are not linguistically robust, and (b) fusion modules are the main failure point due to biases from low linguistic diversity in training data.

**Strengths:**

- Clean dataset construction: the 3D-LR benchmark is well verified (BERT/Glove similarity > 0.9, human Fleiss’ κ≈ 0.56–0.70).
- Simple, practical solution: the pre-alignment module genuinely improves robustness without retraining.
- Readable and polished: figures and supplementary results are professional and exhaustive.

**Weaknesses:**

- The paper repackages a straightforward prompt-based text normalizer as an “LLM-driven pre-alignment module” as this is a quite standard text to diffusion models to align human input and training distribution.
- Limited linguistic realism. The five defined “language styles” (syntax, voice, modifier, accent, tone) are handcrafted and English-centric. I think more discussion about pragmatic. Pragmatic competence is the missing piece for true language understanding in 3D embodied agents.  3D-LR only tests whether models trained on narrow linguistic distributions can handle rewordings. That’s language robustness, not pragmatic comprehension.
- Interpretation drift. The framing question “Can 3D-VL models truly understand language?” is philosophical; the paper only measures paraphrase robustness — a much narrower notion of “understanding.”

**Questions:**

- I wonder what causes 3D LLM this issue. Is this training data? as 3D LLM training data is small? Or is this related to base models? Or is this related to SFT?
- Can the pre-alignment module generalize to non-English languages?
- How do the results compare to simply fine-tuning models with light textual augmentation (back-translation or paraphrasing) rather than LLM pre-alignment?

---

### Official Review · Reviewer_GXvU · 2025-11-01

**Soundness:** 2
**Presentation:** 1
**Contribution:** 2
**Rating:** 2
**Confidence:** 5

**Summary:**

Based on the finding that existing 3D VLMs are not robust to variations of language inputs, this paper proposes a 3D language robustness dataset (3D-LR), which provides diverse language styles to mimic in-the-wild human language. The styles include five most common variants in human communications: syntax, voice, modifier, accent, and tone. Based on the proposed dataset, the authors evaluate a variety of 3D VLMs and reveal their weak robustness when encountering language variations. Furthermore, this paper proposes a simple module to address the robustness issue for 3D VLMs without additional training.

**Strengths:**

- The robustness issue appears to be a profound problem for current 3D VLMs. This paper focuses on this significant problem and proposes a targeted dataset (3D-LR) with detailed analyses.
- Beyond the findings that current 3D VLMs struggle in language robustness, this paper further explores the reason and proposes a solution to mitigate the robustness issue. This part provides some useful discussion.

**Weaknesses:**

- The finding that 3D VLMs degrade on 3D-LR datasets is somewhat insufficient. I think it lacks a general metric that measures the degradation for a model, which can facilitate the comparison across models. Given the various model baselines, the general metric can also help identify what kinds of models show better robustness.
- I think the analyses in Sec. 5.1 is not convincing enough as the results are not significant. And the proposed pre-alignment module (Sec. 5.2) is not a fundamental solution because it just forces the input to be similar to the training corpus, just like prompt engineering.
- The literature review is out of date. Some recent related works are not properly discussed, including progress in both 3D VLMs and related robustness analyses.

**Questions:**

- Format issue. The page number and header of Page 3 and 4 are inappropriately boxed with red and green colors.
- In Figure 2, I cannot observe a notable difference in the distribution pattern of the four datasets. I wonder of what this can be an evidence.

---

### Note · Authors · 2025-11-14

**Comment:**

We thank the reviewers for their time and comments. We have decided to withdraw the submission and will use the feedback to guide our revisions.

**Withdrawal Confirmation:**

I have read and agree with the venue's withdrawal policy on behalf of myself and my co-authors.